# Red Mud Potentially Alleviates Ammonia Nitrogen Inhibition in Swine Manure Anaerobic Digestion by Enhancing Phage-Mediated Ammonia Assimilation

**DOI:** 10.3390/microorganisms13030690

**Published:** 2025-03-19

**Authors:** Yulong Peng, Luhua Jiang, Junzhao Wu, Jiejie Yang, Ziwen Guo, Manjun Miao, Zhiyuan Peng, Meng Chang, Bo Miao, Hongwei Liu, Yili Liang, Huaqun Yin, Qiang He, Xueduan Liu

**Affiliations:** 1School of Minerals Processing and Bioengineering, Central South University, Changsha 410083, China; 225611009@csu.edu.cn (Y.P.); 245612148@csu.edu.cn (J.W.); jiejieyang212@foxmail.com (J.Y.); kidgzw@hotmail.com (Z.G.); 235612083@csu.edu.cn (M.M.); 225611007@csu.edu.cn (Z.P.); 8202221031@csu.edu.cn (M.C.); miaobo@csu.edu.cn (B.M.); hongweiliu@csu.edu.cn (H.L.); liangyili6@csu.edu.cn (Y.L.); yinhuaqun_cs@sina.com (H.Y.); xueduanliu@csu.edu.cn (X.L.); 2Key Laboratory of Biometallurgy of Ministry of Education, Central South University, Changsha 410083, China; 3Department of Civil and Environmental Engineering, University of Tennessee, Knoxville, TN 37996, USA; qianghe@utk.edu

**Keywords:** anaerobic digestion, red mud, phage, microbiomics, ammonia assimilation, swine manure

## Abstract

Red mud has been demonstrated to improve the methane production performance of anaerobic digestion (AD). However, the influence of red mud on ammonia nitrogen inhibition during AD through the mediating role of bacteria–phages interactions in this process remains poorly understood. Thus, this study investigated the impact of red mud on nitrogen metabolism in AD and characterized the phage and prokaryotic communities through a metagenomic analysis. The results showed that red mud significantly increased methane production by 23.1% and promoted the conversion of ammonia nitrogen into organic nitrogen, resulting in a 4.8% increase in total nitrogen. Simultaneously, it enriched the key microbial genera *Methanothrix*, *Proteinophilum*, and *Petrimonas* by 0.5%, 0.8%, and 2.7%, respectively, suggesting an enhancement in syntrophic acetate oxidation with greater ammonia tolerance. A viral metagenomic analysis identified seven nitrogen-metabolism-related auxiliary metabolic genes (AMGs), with *glnA* (encoding glutamine synthetase) being the most abundant. Compared to the control treatments, the red mud treatments led to a higher abundance of temperate phages and an increased number of AMGs. Furthermore, two new hosts carrying *glnA* (*Mycolicibacteria smegmatis* and *Kitasatopola aureofaciens*) were predicted, indicating that red mud expanded the host range of phages and promoted the spread of AMGs. Overall, our findings highlight the importance of phages in alleviating ammonia nitrogen inhibition and provide a novel understanding of the role of red mud in the AD of swine manure.

## 1. Introduction

The anaerobic digestion (AD) of swine manure is a widespread and environmentally sustainable treatment practice, efficiently transforming the manure into renewable fuels and organic fertilizers, thus facilitating resource recycling and conservation [1,2,3]. However, the degradation of urea and proteins in swine manure usually results in excessive levels of ammonia nitrogen, which inhibits the AD process [4,5,6]. The mechanism of ammonia nitrogen inhibition involves ammonia causing a proton imbalance and potassium loss, which, in turn, affects microbial metabolism [7,8,9]. Moreover, ammonia can also directly inhibit enzymes involved in methane production [4]. Ammonia nitrogen inhibition has led to a reduction in methane production by up to 30% in numerous anaerobic reactors, causing substantial economic losses [10]. As a consequence, alleviating ammonia nitrogen inhibition is the key to improving the efficiency of the AD of swine manure.

Red mud, also known as bauxite residue, is a byproduct resulting from the alumina refining process of bauxite ore via the Bayer method. Its main components are Fe_2_O_3_, Al_2_O_3_, SiO_2_, and a small amount of rare metal oxides. [11]. According to statistics, the global annual production of red mud exceeds 150 million tons, yet its comprehensive utilization rate is less than 10% [12]. Recently, studies have continuously reported its superior performance in AD. For example, the hydrolysis–acidification and electron transfer processes were improved by red mud with the simultaneous enhancement of methanogenesis [13,14]. Furthermore, the micro- and macronutrients in red mud, such as manganese and sodium, were beneficial for the growth of microorganisms [13]. Notably, because of the high content of nitrogenous compounds in swine manure, excessively rapid hydrolysis–acidification will lead to the significant production of ammonia nitrogen, which may cause ammonia nitrogen inhibition and limit the application of red mud in AD. Although there is currently a lack of evidence to indicate that red mud affects nitrogen metabolism in AD, some researchers have found that poultry manure can improve the nitrogen availability and microbial functionality of bauxite residue as growth media for plants [15]. Hence, it is inferred that the incorporation of red mud has a potential impact on microbial-mediated nitrogen cycling during AD processes. Moreover, the research into the genetic-level responses of microorganisms to red mud remains insufficient.

Phages, viruses that infect bacteria, are abundant in AD systems and regulate the nitrogen cycle and methanogenesis in various ways [16]. Phages have the capability to lyse microbial cells, thereby releasing nutrients derived from the host and drive the viral shunt [17]. In addition, phages also participate in the nitrogen cycle by expressing auxiliary metabolic genes (AMGs) to modulate host nitrogen metabolic pathways, such as nitrification, denitrification, anammox, and nitrogen transmembrane transport [18,19]. Numerous studies have revealed the effects of viruses on hosts under the modulation of environmental factors [20,21]. For example, hematite was found to increase phage infectivity in an abandoned mine [22]. The multivalent cations from hematite effectively promoted the formation of large and compact aggregates, which might contribute to the spread of bacteriophages between different hosts [13,23].

Therefore, we have compelling reasons to hypothesize that phages may play a pivotal yet underappreciated role in ammonia nitrogen inhibition, and red mud may have an influence on the functional capabilities of phages. In this study, red mud was added to the batch mode AD system of swine manure to alleviate ammonia nitrogen inhibition. The viral community and other microbial communities were analyzed through metagenomics. The present study aims to unveil the mechanism of red mud enhancing methane production in swine AD from a microbial perspective, determine the effectiveness of red mud in alleviating ammonia nitrogen inhibition, and compare the changes in the functions of viruses in ammonia nitrogen metabolism and methane production before and after adding red mud.

## 2. Materials and Methods

### 2.1. Materials

Bayer red mud was provided by the Aluminum Corporation of China, Jiaozuo, Henan, China [24]. The mineral compositions of red mud are provided in Appendix A. The seed sludge was obtained from a sewage-treatment plant of Changsha Drainage Co., Ltd., located in Changsha, Hunan, China. The sludge, taken from a secondary sedimentation tank, was thickened by gravity settling and stored at 4 °C as inoculum. The liquid fraction of swine manure, obtained following a dry–wet separation process at a pig farm in Liuyang, Hunan, China was utilized as the substrate for this research. Subsequently, both the inoculum and swine manure were subjected to comprehensive characterization, which included assessments of total solids (TS), volatile solids (VS), and soluble chemical oxygen demand (SCOD) with standard methods [25]. An overview of the main information is provided in Appendix A.

### 2.2. Anaerobic Digestion Experiments

The batch experiment was performed in 250 mL anaerobic serum bottles, filled with 10 mL inoculum and 90 mL swine manure [14]. Based on the results of the pre-experiment (Appendix A), the test bottles that contained 0.5% red mud were labeled RM, and the control bottles without red mud were labeled CK. After that, the bottles were subjected to a rigorous anaerobic environment by purging the headspace with nitrogen gas for a duration of 15 min [26]. The entire experiment was conducted in a constant-temperature incubator set at 38 °C with a shaking speed of 180 rpm. The cessation of gas production signified the end of the experiment, and all treatments were performed in triplicate [27].

### 2.3. Analytical Methods

We utilized the gas chromatograph (GC-7900, Agilent, Santa Clara, CA, USA), which was equipped with a thermal conductivity detector and employed helium as the carrier gas at a temperature of 170 °C, to quantify the daily gas composition, including H_2_, N_2_, CO_2_, and CH_4_. The cumulative volumetric production of biogas was determined using the water displacement method. By introducing the gas into a U-shaped glass tube filled with water, the biogas production could be calculated based on the volume of water displaced. Ultimately, methane production was calculated based on the CH_4_ gas composition and the total biogas production. To assess the influence of red mud on nitrogen metabolism, the concentrations of ammonia nitrogen, nitrate nitrogen (NO_3_^−^), nitrite nitrogen (NO_2_^−^), and total nitrogen were systematically monitored at intervals of three days. The concentration of ammonia nitrogen was determined using Nessler’s reagent with a spectrophotometer (UV-1780, Shimadzu, Kyoto, Japan). The NO_3_^−^ and NO_2_^−^ concentrations were measured through the reagents LH-NO3 and LH-NO2 (Lianhua Technology, Shenzhen, China), respectively, with UV-1780 (Shimadzu, Japan). A total nitrogen analyzer (LH-TN360, Lianhua Technology, China) was utilized to determine the TN concentration. The red mud sample was dried at 105 °C until a constant weight was achieved, ground into a fine and homogeneous powder, and then passed through a 200-mesh sieve. Then, the mineral compositions of red mud were detected by X-ray diffraction (XRD-6000, Shimadzu, Japan). To investigate whether red mud promoted the formation of microbial aggregates, the sediments of AD were assessed using a scanning electron microscope (SEM) (CUBE-1100, Emcrafts, Gwangju, Republic of Korea) equipped with an energy-dispersive spectrometer (EDS) (Xplore 30, Oxford Instruments, Abingdon on Thames, UK). The HV was set to 20 kV, the MAG was 5.00 kx, and the WD was 12.95 mm.

### 2.4. Viral DNA Extraction and Sequencing

According to the manufacturer’s protocol, all samples were transferred into sterilized bottles and centrifuged, although this process might result in some loss of virus [28,29]. Viral nucleic acids were extracted from the samples through a viral extraction kit, followed by whole-genome amplification by using an Illustra Ready-To-Go GenomiPhi V3 DNA Amplification Kit [30]. The resulting amplified products were then quality-checked using a Thermo NanoDrop One, Life Technologies Qubit 4.0, and 1.5% agarose gel electrophoresis. The extracted viral DNA was sequenced on an Illumina Novaseq X plus PE150 platform [31]. To acquire clean data for subsequent analysis, the raw data were first screened with FastQC (v0.23.4) to assess quality, followed by the trimming and filtering of low-quality reads using Trimmomatic (v0.40) [32,33]. The extraction and sequencing methods of total DNA are described in Appendix A. Sequencing data were deposited in the NCBI (National Center for Biotechnology Information) Sequence Read Archive database with accession number PRJNA1207431.

### 2.5. Viral Contig Assembly, Identification, and Host Prediction

Decontaminated clean reads were assembled through the Megahit software (version v1.2.9, default parameters: mem-k30) [34]. Subsequently, clean reads were aligned to the host using the BWA software (version 0.7.17) to mitigate the influence of host sequences on subsequent analyses. Alignments with a length shorter than 80% of the total read length or similarity ≥90% were filtered out, and the corresponding sequences were subsequently removed [35]. Potential viral contigs in assembled sequences were annotated using the “Prodigal” script in CheckV (v0.8.1) [36]. Meanwhile, the annotated genes were compared with the Hidden Markov Model constructed from the virus database V3 built by Megigene, to identify the genes of viruses and other microorganisms. The initially assembled sequences underwent additional classification utilizing the Virsorter2 software (version v2.2.3), thereby reinforcing the conclusions derived from CheckV. Regarding the proviral sequence, it is imperative to excise the host-derived portion [37]. Subsequently, the viral sequences identified for each sample underwent clustering and redundancy removal, with a threshold of >95% identity and >80% coverage, utilizing the PSI-CD-HIT script within the CD-HIT tool (version v4.8.1) [31]. Following this process, the generation of “viral populations” (viral operational taxonomic units, vOTUs) was successfully performed [38]. The longest contig within each viral cluster (VC) was designated as the representative sequence for subsequent taxonomic analysis and classification [31]. Clean reads post-pollution removal were compared with viral contigs using the BWA software (v 0.7.17). Results with lengths < 80% of the total reads were filtered, and the RPKM values were calculated for each viral contig [39].

The identified viral sequences were analyzed for host prediction using the CHERRY software (version v2023) and the PHP software (version v2023), followed by the statistical compilation of host species information based on the analysis outcomes [40,41]. CHERRY is a piece of software based on a deep learning model consisting of a graph convolutional encoder and a link prediction decoder. It possesses the capability to predict the hosts of viruses at the species level. PHP software is a prokaryotic virus host prediction tool based on a Gaussian model. High-reliability host prediction results are obtained by calculating the host probabilities of prokaryotic genomes and setting a threshold for matching scores.

### 2.6. Taxonomic Assignment and AMG Annotation of Phages

After successfully acquiring the aforementioned presumptive collections of viral sequences, the phages underwent identification using the PhaMer software (v 1.0) [42]. Subsequently, taxonomic annotation and lifestyle prediction of these phages were executed by employing the PhaGCN2 software (version v2.0) and PhaTYP software (v 1.0) [43,44]. By integrating the annotation outcomes from PhaGCN2 with the annotated information of target alignment sequences in the databases, the species information of the phage sequences was jointly confirmed. Ultimately, the phage sequences were classified according to the identification method, confidence level, and completeness information. Comparative analyses were then conducted among different samples. The functional annotation of vOTUs involved assigning predicted proteins to the KEGG database and NcycDB database [45]. Therefore, the obtained functional annotations were manually managed to screen out ammonia-nitrogen-metabolism-related AMGs.

### 2.7. Statistical Analysis

All statistics analyses were performed using the R software (3.5.3 version). An ɑ-diversity nonmetric and multidimensional scaling (NMDS) analysis and a multivariate analysis of similarity (ANOSIM) were performed using the “vegan” package (v 2.6-4). The differentially abundant vOTUs among treatments were identified using the linear discriminant analysis (LDA) effect size (LEfSe) method (*p* < 0.01 and an LDA score >2.5) [46,47]. The co-occurrence networks were drawn using Gephi (v 0.10) [48]. Additional charts were generated utilizing the Origin 2024b software [9].

## 3. Results and Discussion

### 3.1. Impacts of Red Mud on Methane Production and Nitrogen Metabolism

All batch experiments were performed over 15 days after adding red mud. The cumulative methane yield is shown in Figure 1a. After 12 days, the methane yield tended to be stopped, and the methane yield of the RM was 143.08 ± 4.49 L CH_4_/kg VS. In comparison to CK, the cumulative methane yield in RM was significantly increased by ~23.1%. This indicated that red mud had a significant promoting effect on methane production, which was consistent with the results reported by Zhou et al. [13]. Volatile fatty acids (VFAs), with acetic acid in particular, serve as key precursors for the synthesis of methane. Notably, obligate acetotrophic methanogens, which utilize acetic acid as a nutrient substrate, are particularly vulnerable to ammonia toxicity [49]. Previous research indicated that red mud increased the content of VFAs in anaerobic conditions when BESA was added to inhibit methane generation [12]. Consequently, the concentrations of VFAs were measured on the third day (hydrolysis–acidification process) in this experiment with the absence of BESA. As shown in Figure 1b, based on the results of methane generation, red mud promoted the consumption of VFAs. These results indicated that the activity of acetotrophic methanogens might have been enhanced.

In the AD process, ammonia nitrogen is produced through the hydrolysis and acidification of the nitrogenous matter in the swine manure. Investigating the effect of red mud on ammonia nitrogen is essential for maintaining the stability of AD. Figure 1c shows the effect of red mud on nitrogen metabolism in AD. As anticipated, the facilitation of protein hydrolysis through the red mud resulted in the augmentation of various forms of nitrogen, predominantly ammonia nitrogen, accompanied by a negligible amount of NO_3_^−^ and undetectable NO_2_^−^. However, the concentration of ammonia nitrogen in the RM amounted to 563.28 mg/L, markedly lower than that of CK, which stood at 631.22 mg/L. Meanwhile, the concentration of total nitrogen in RM was recorded to be 781.42 mg/L, representing an increase of 4.8% when compared to CK. These findings suggest that the red mud could potentially result in greater conversion of ammonia nitrogen into organic nitrogen.

### 3.2. Community Structure and Diversity of Microbes

The stability within AD systems is frequently mirrored by the microbial structure and diversity. During the AD processes, the bacterial community underwent significant species turnover. The composition and abundance of the microbial community structure at the phylum and genus taxonomic levels are shown in Figure 2a,b. After the introduction of red mud, the microbial communities within the AD system underwent significant species turnover. As shown in Figure 2a, Proteobacteria, Bacteroidetes, and Firmicutes were the dominant phyla. Bacteroidetes is a well-known phylum that participates in the breakdown of long-chain polymers and the production of VFAs [50]. Notably, its relative abundance in RM increased by 7% compared to CK. The enrichment of Bacteroidetes corresponds well to a higher VFA concentration. The relative abundance of Firmicutes exhibited an upward trend, with a 0.7% increase in RM compared to CK. Firmicutes plays a pivotal role in the metabolism of cellulose, proteins, lignin, and lipids through the secretion of extracellular enzymes. Certain studies have observed a positive correlation between the stability of the AD process and the abundance of Firmicutes populations [51,52].

The findings presented in Figure 2b reveal a change in the dominant genus caused by red mud. The relative abundances of *Methanothrix*, *Proteinophilum*, and *Petrimonas* were significantly increased by 0.5%, 0.8%, and 2.7%, respectively, following the red mud treatment. *Petrimonas* could potentially enhance the optimization of VFA composition and expedite methane production by promoting the formation of acetic acid and carbon dioxide. Furthermore, *Petrimonas* exhibited notable adaptability in response to corresponding stress environments and prevailed in competition with other bacteria that exhibited lower stress resistance [53]. *Proteiniphilum* are likely to function as acetate oxidizers within the AD system, as they exhibit high expression of genes responsible for syntrophic acetate oxidation, hydrogen production, and electron transfer processes [54]. Methanogenic archaea of the genus *Methanothrix* are known to play a crucial role in maintaining stable ecosystem functioning within anaerobic bioreactors [55]. Notably, *Proteiniphilum* as well as *Methanothrix* exhibited strong ammonia tolerance. Under ammonia nitrogen inhibition conditions, the syntrophic acetate oxidation (SAO) induced by these two microorganisms plays a crucial role in the conversion of acetic acid to methane [56]. This result indicates that with the addition of red mud, SAO gradually became the main pathway for methane generation.

The alpha-diversity indexes of microbial communities at the genus level are summarized in Appendix A. Compared with CK, various diversity indices were lower following the red mud treatments. A reasonable interpretation of this occurrence is that the addition of red mud introduces a range of metal ions into the environment and raises the pH, which then suppresses the activity of certain microorganisms and aids in the emergence of dominant populations. Nevertheless, some studies have revealed that this phenomenon does not invariably have a detrimental effect on AD. For example, an elevated initial sludge pH level effectively hindered the competition between sulfate-reducing bacteria and methane-producing bacteria, ultimately fostering the proliferation of methanogens [57]. In addition, calcium ion contained in red mud could alleviate ammonia nitrogen inhibition in AD by balancing and strengthening dehydrogenases and reinforcing protein-binding structure [58].

NMDS analysis based on the Bray–Curtis distance revealed the dissimilarity of microbial communities before and after different treatments (Figure 3a). These results showed significant disparities in the microbial community structure between the initial and treated samples. This is also corroborated by a Venn diagram of shared or endemic species composition (Figure 3b), indicating a larger number of shared species between CK and RM. And compared with CK, the three parallel samples of RM exhibited smaller variations. This is because after AD, as the oxygen levels decreased, the microbial community structure underwent notable alterations. Furthermore, considering the change in alpha-diversity, this was reasonable considering that red mud promoted the establishment of dominant populations, resulting in species similarity exceeding that of CK.

The results obtained through SEM-EDS (Figure 4) intuitively demonstrated the impact of red mud on microbial community structure. The red mud particles rich in iron and aluminum served as conductive solid conduits, improving electron transfer efficiency and promoting the formation of microbial aggregates [13]. Therefore, methane production pathways that were more dependent on electron transfer and microbial collaboration, such as syntrophic acetate oxidation, were enhanced.

### 3.3. Composition and Evolution of Viral Community

Viral metagenomic sequencing generated 176 million paired-end reads, then assembled them into 2.55 million contigs, among which 172,656 contigs were identified as viral genomes (Appendix A). The low proportion of viral contigs might have been caused by the high cell density of AD sludge and the lack of DNase I treatment. The high cell density might have hindered the efficiency of virus-like particle enrichment, and without DNase I treatment, the extracellular DNA of prokaryotes may have occupied some of the sequencing data volume, resulting in the low proportion of viral contigs in this study. A total of 76.89% of viruses possessed dsDNA, and the population of ssDNA viruses was significantly low, which was analogous to the situations found in most AD reactors (Appendix A).

As listed in Appendix A, compared with CK, the diversity indexes were higher in RM. NMDS analysis revealed dissimilarity of the viral communities between different treatments on (Figure 5a). Following the latest ICTV taxonomy database, viral communities were assigned to Caudoviricetes, Faserviricetes, DArfiviricetes, Megaviricetes, and Malgrandaviricetes at the class level, among which Caudoviricetes accounted for more than 53% of the relative abundance (Figure 5b). Tailed phages (*Caudovirales*) constitute over 90% of all phages documented thus far and exhibit an extraordinary level of diversity in AD systems. Recent studies have shown that Caudoviricetes can host methanogenic archaea and facilitate horizontal gene transfer [59,60]. Therefore, the increase in relative abundance might reflect the enhanced influence of phages on AD processes. The LEfSe analysis identified nine families (*p* < 0.05, LDA score > 2) that showed significantly different relative abundances between treatments (Figure 5c). *Inoviridae* was the most differentially abundant family in RM, while *Circoviridae* and *Genomoviridae* were the most differentially abundant families in CK. All of these families are members of the ssDNA virus. However, the abundance of ssDNA viruses, represented by the aforementioned families, may have been overstated through the use of MDA amplification in the present study [61].

Employing viral contigs as inputs for PhaMer, a deep learning-based model that utilizes protein clusters as tokens for phage identification, 3359 and 3340 contigs were identified as phages in the CK and RM datasets, respectively [42]. A total of 2118 virulent phages and 1241 temperate phages were further identified in CK, whereas 2065 virulent phages and 1275 temperate phages were identified in RM (Figure 6a). Notably, a higher relative abundance of temperate phages was observed in RM. Virulent phages follow the “kill the winner” model, which means they suppress dominant bacteria in the environment, increase the frequency of minority groups, and maintain the diversity of bacterial communities. Thus, the reduction in the proportion of virulent phages aligned with the decline in microbial diversity [62]. Moreover, phages usually exhibit a high lysogenic proportion when exposed to harsh conditions, such as those found in heavily polluted soil, arid deserts, glaciers, and oceans [63,64,65]. An increase in lysogenic conversion rates might increase the frequency of horizontal gene transfer in microbial communities via specialized transduction and lateral transduction, [66,67] which is beneficial for the recovery of microbial adaptability under high ammonia nitrogen stress.

The co-occurrence networks of different treatment groups for the phage communities are depicted in Figure 6c. The results indicate that there was no significant difference in the complexity of phage networks, possibly because the pressure of red mud was insufficient to disrupt the individual structure and physiological processes of phages [68]. The phage classification is shown in Figure 6b; about 80% of contigs could not be classified at the family level. *Straboviridae*, *Peduoviridae*, and *Zierdtviridae* are dominant families in both treatments, and the relative abundance of *Drexlerviridae* significantly increased in RM. Corresponding to the results in Figure 5b, *Drexlerviridae* is a family under the class *Caudoviricetes*, but its function in AD processes remains to be studied. Therefore, it is necessary to pay attention to *Drexlerviridae* in AD-related research in the future.

### 3.4. Phage-Encoded AMGs Enhanced Anaerobic Digestion and Ammonia Assimilation

AMGs can either sustain, manipulate, or commandeer the metabolic pathways of their hosts post-infection, offering significant adaptive benefits to phages in particular environmental conditions, and potentially influencing the assimilation of the host and the breakdown of organic compounds [69]. Through the investigation of AMGs associated with acidification, methane metabolism, and nitrogen metabolism during AD processes, we elucidated the mechanism by which red mud can augment host ammonia tolerance and enhance AD processes by having an influence on phages. As depicted in Figure 7a, the introduction of red mud led to a substantial rise in all categories of AMGs, particularly those associated with amino sugar and nucleotide sugar metabolism, as well as pyrimidine metabolism. The quantity of contigs encoding these pathways amounted to 825 and 1336, respectively. Energy is crucial to the physiological functions of microbes and is generated via the process of acidification. A strategy of encoding acidification-related AMGs might be employed by phages to acquire additional reproductive energy, which could be vital for their survival in hostile environments. Hence, phages could promote the acidification of organic matter, thereby facilitating the production of intermediate products (such as VFAs and ammonia) and contributing to the subsequent methanogenesis process. However, the significant quantities of ammonia nitrogen generated through amino acid acidification might potentially suppress subsequent methane production.

To further elucidate the fate of ammonia nitrogen generated through acidification, the AMGs associated with nitrogen metabolism were studied in depth. Seven AMGs associated with nitrogen metabolism were identified on 140 contigs in RM and 81 contigs in CK (Appendix A). We found that phages carrying key glutamine synthetase (glnA) higher coding abundance in RM (Figure 7b). Glutamine synthetase, as a fundamental enzyme in the process of ammonium assimilation, catalyzes the condensation reaction between ammonia and glutamic acid to ultimately yield glutamine, thereby alleviating the stress of ammonia nitrogen on microbial communities [70]. Consequently, phages might alleviate the inhibition of ammonia by encoding the glnA gene. In conclusion, phages encoded a greater number of AMGs to fulfill the requirements for survival and proliferation in the presence of red mud stress, encompassing acidification and ammonia assimilation, which likely enhance the ammonia nitrogen metabolism and adaptability of the host.

### 3.5. Phage–Host Linkage Analyses

Phages modulate the composition and metabolic activities of their host communities via various top-down mechanisms, including host lysis and metabolic assistance. We only predicted the hosts of phages carrying nitrogen-metabolism-related AMGs to reveal the potential impact of phages on ammonia nitrogen inhibition from the host perspective (Figure 8a). Notably, due to the fact that the majority of annotated AMGs had too short sequences (less than 3000 kb in length), these sequences were filtered during host prediction, making it difficult to predict the lifestyle and hosts of phages carrying these AMGs using existing tools. In the process of nitrogen metabolism, the phage–host interactions associated with glutamine synthesis were particularly prevalent. A total of 107 glnA genes were found to be carried by phages from the Caudoviricetes class after adding red mud. In comparison to CK, where only *Candidatus Hamiltonella defensa* was predicted, the hosts in RM also included *Mycolicibacterium smegmatis* and *Kitasatospora aureofaciens*. Interestingly, *Candidatus Hamiltonella defensa* possesses the capability to utilize the nitrogenous waste products of its insect host to synthesize essential amino acids and B vitamins [71]. Conversely, there have been no reports of *Mycolicibacteria smegmatis* and *Kitasatopola aureofaciens* making significant contributions to nitrogen metabolism during AD processes.

In the present study, the patterns of co-occurrence between phages and bacteria at the species level were evaluated through a network analysis approach (Figure 8b). A comprehensive coexistence network encompassing all samples was established for phages and bacteria, and the topological properties of the network are shown in Appendix A. Compared with the CK network, despite the RM networks having fewer nodes and edges, other topological properties were observed to increase. These networks could be categorized into numerous primary modules, based on the clustering characteristics of the nodes. The nodes that were most densely connected within each module were referred to as hubs, which were primarily from *Mycobacterium smegmatis* and *Escherichia coli*. It is worth noting that *Mycolicibacteria smegma* served as the host for phages that emerged in RM, carrying the glnA genes (Figure 8a). These findings suggest that red mud enhanced the interactions between bacteria and phages, resulting in a more intricate bacteria–phage ecological network. Considering that phage transduction entails the infection of both the bacterial donor and the recipient, the intricate and closer links between bacteria and virus might play an pivotal role in facilitating the exchange of AMGs between them. The density of hosts is a key parameter that determine the host range. Intuitively, expanding the host range is advantageous for phages because they can infect more hosts. However, infecting new hosts requires more energy [62]. Overall, red mud was able to enrich bacterial populations and facilitate the establishment of dominant species, thereby augmenting host density. In addition, it also expedited the hydrolysis and acidification processes, ultimately enhancing the availability of nutrients for phage reproduction. Moreover, although the metaviromic data suggest that bacteriophages play a significant role in alleviating ammonia inhibition, direct evidence supporting these findings cannot be provided due to the lack of further investigation into processes such as transcription and expression. Therefore, additional work, including metatranscriptomic and metaproteomic analyses, is needed for more in-depth research and exploration.

## 4. Conclusions

In summary, this study provides novel insights into the role of red mud in AD, as well as the diversity and horizontal transfer of nitrogen-metabolism-associated AMGs carried by phages in the AD process. Red mud enriched functional microorganisms related to syntrophic acetate oxidation, thereby accelerating the conversion of VFAs into methane, which was beneficial for methane production under ammonia nitrogen stress. In RM, more *glnA* genes were carried by phages, indicating an enhanced ammonia assimilation process, which led to a decrease in ammonia nitrogen concentration in the AD process. Additionally, the increased proportion of lysogenic phages and the expansion of their host range suggested a higher frequency of horizontal gene transfer, underscoring the importance of phages in mitigating ammonia nitrogen inhibition. This study sheds light on the potential mechanism through which red mud alleviates ammonia nitrogen inhibition via phages, providing a theoretical foundation for the utilization of red mud in AD under high ammonia nitrogen conditions.

## Figures and Tables

**Figure 1 microorganisms-13-00690-f001:**
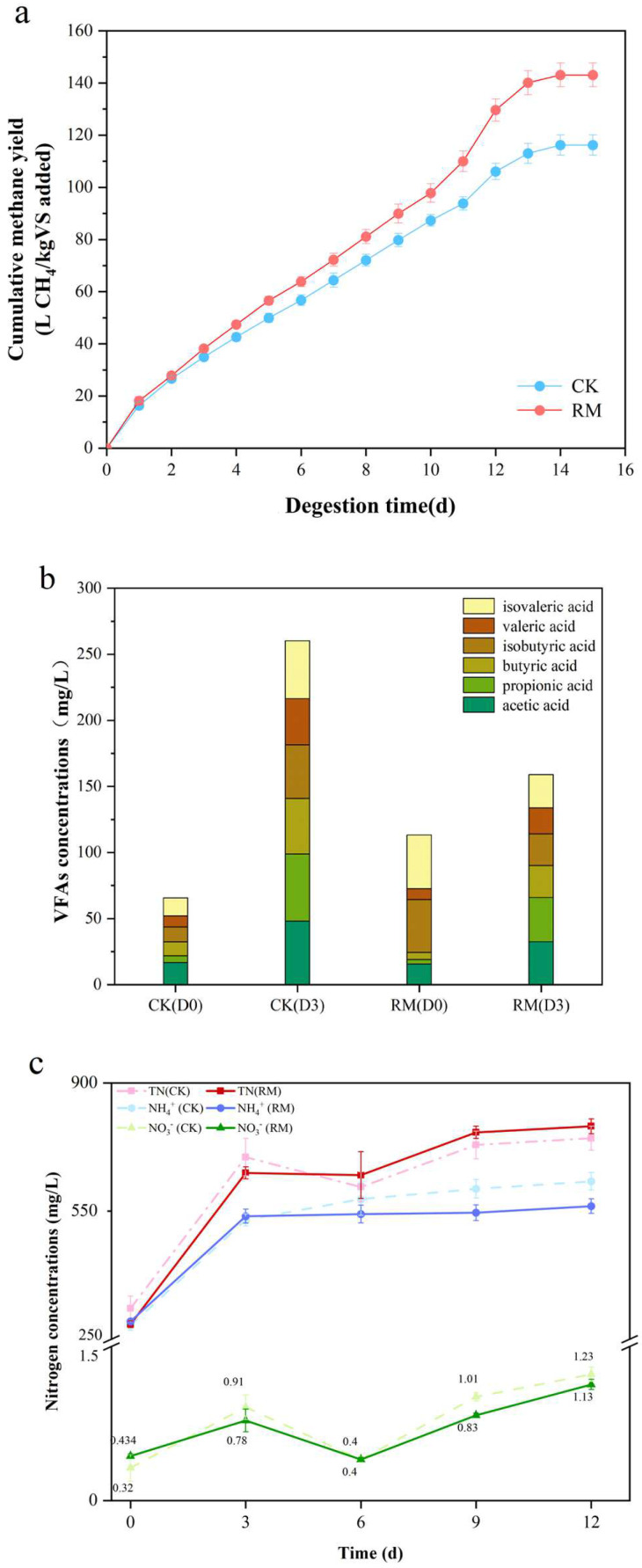
The impact of red mud on AD. (**a**) The production of methane in the batch experiments. (**b**) The concentrations of VFAs, including acetic acid, propionic acid, butyric acid, isobutyric acid valeric acid, and isovaleric acid. (**c**) The changes in nitrogen content in different forms during AD processes.

**Figure 2 microorganisms-13-00690-f002:**
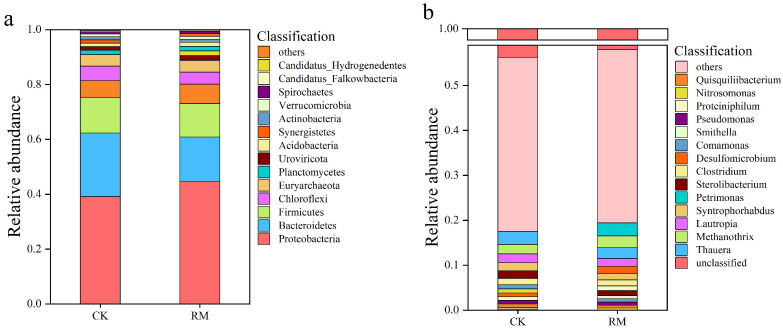
Community structure of microbes in CK and RM. Relative abundance of dominant microbial groups at phylum (**a**) and genus (**b**) levels.

**Figure 3 microorganisms-13-00690-f003:**
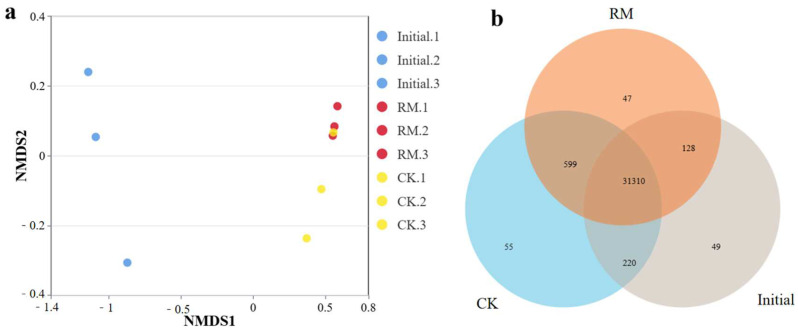
Diversity and differentiation of microbial community between different treatments. (**a**) NMDS analysis of microbial community compositions based on Bray−Curtis similarities. (**b**) Venn diagram of shared or endemic species composition.

**Figure 4 microorganisms-13-00690-f004:**
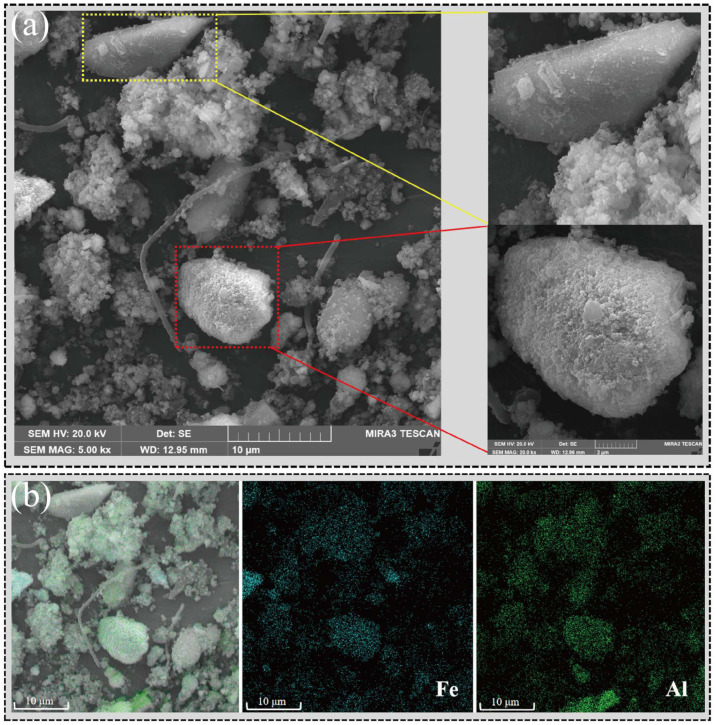
Red mud particles promoted the formation of microbial aggregates. (**a**) SEM results from RM. (**b**) EDS analysis results from RM. Blue symbolizes iron element, whereas green signifies aluminum element.

**Figure 5 microorganisms-13-00690-f005:**
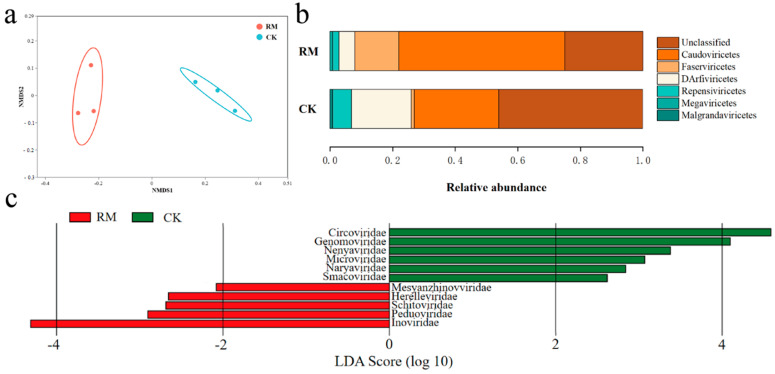
Diversity and differentiation of microbial community between different treatments. (**a**) NMDS analysis of viral community compositions based on Bray−Curtis similarities. (**b**) Relative abundance of dominant virus at class level. (**c**) LEfSe analysis showing differentially abundant viruses at family level. Results based on *p* < 0.05 and LDA score > 2.

**Figure 6 microorganisms-13-00690-f006:**
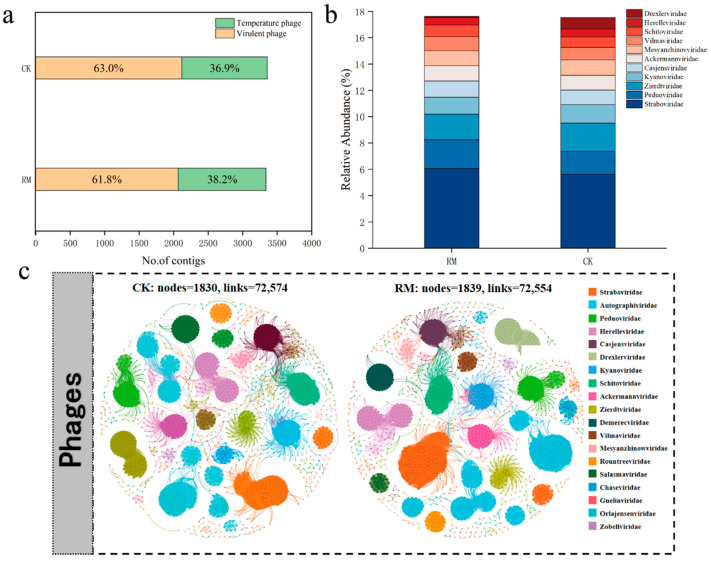
Identification and distribution of phages. (**a**) Number of lytic and temperate phage contigs identified. (**b**) Relative abundance of top 10 most abundant phage families. (**c**) Co-occurrence network of phage communities in CK and RM.

**Figure 7 microorganisms-13-00690-f007:**
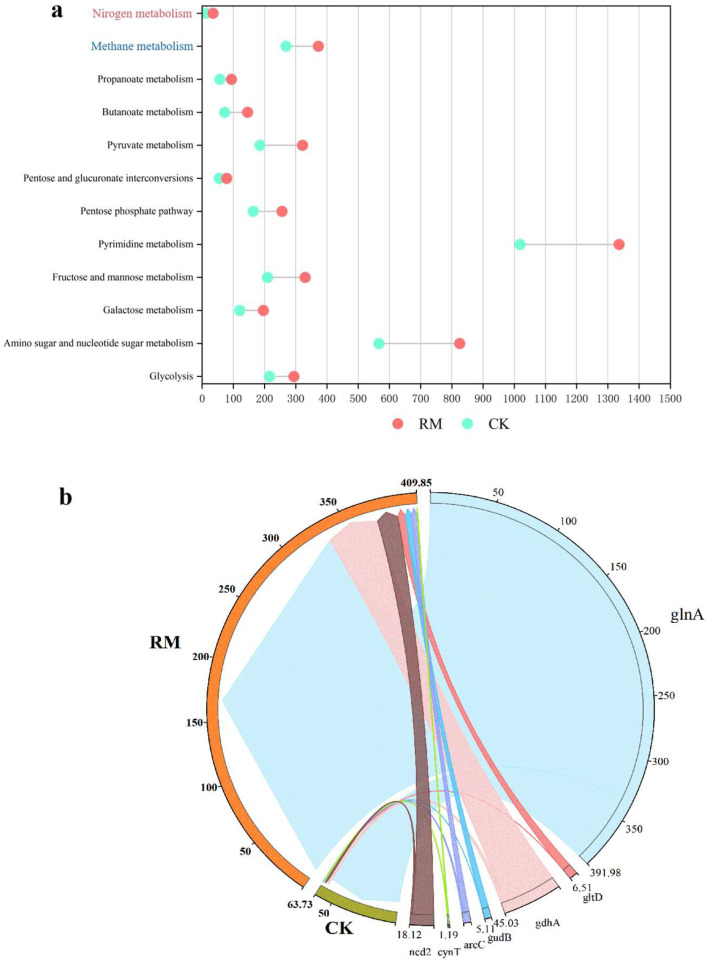
AMGs in AD digesters. (**a**) Distribution of AMGs relevant to methane production and ammonia nitrogen metabolism. (**b**) RPKM of nitrogen metabolism-related AMGs.

**Figure 8 microorganisms-13-00690-f008:**
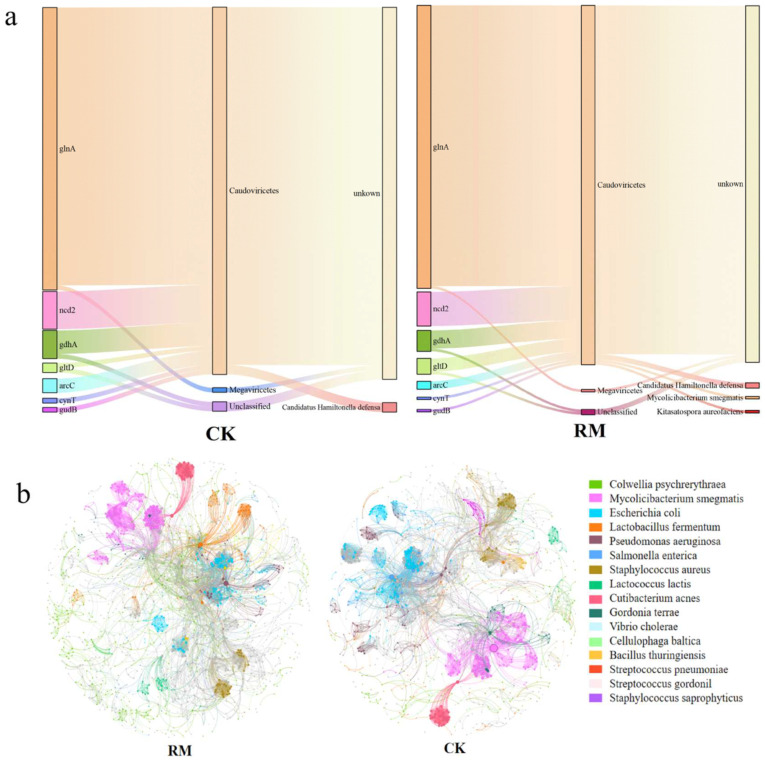
The linkage between phages and hosts. (**a**) The types, phage classes, and hosts of nitrogen-metabolism-related AMGs. (**b**) Co-occurrence networks between phages and bacteria.

## Data Availability

The original contributions presented in this study are included in the article. Further inquiries can be directed to the corresponding author. The metagenomics and viromics data presented in the study are openly available in the SRA Database at NCBI (https://www.ncbi.nlm.nih.gov/sra/SRX27321562), accession number PRJNA1207431, accessed on 16 January 2025.

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
