# Peer review of "Red Mud Potentially Alleviates Ammonia Nitrogen Inhibition in Swine Manure Anaerobic Digestion by Enhancing Phage-Mediated Ammonia Assimilation"

_microorganisms, 2025, doi:10.3390/microorganisms13030690_

Round 1
Reviewer 1 Report
Comments and Suggestions for Authors
This is an interesting article.
To strengthen it, you could provide more context:
How does the total amount of red mud compare to the needs for biogas production? On a domestic and global scale. Can the red mud be reused/recycled in AD?
What about aluminium contamination of the red-mud enriched AD residue? Where to dispose of it? Can it lead to problems? Are there any safety concerns in transporting or storing the red mud?
What about a circular use of aluminium, where less and less primary aluminium is needed?
The title says "potentially"; How sure are you about your conclusions? How could you verify the assumption and advoid the "potentially"?
Minor comments:
199: All The batch experiments --> All batch experiments
Fig. 1: Digetion time -->Digestion time
Reviewer 2 Report
Comments and Suggestions for Authors
The manuscript represents a research paper that aimed to investigate the impact of red mud on nitrogen metabolism in anaerobic digestion of swine manure and characterize the phage and prokaryotic communities through metagenomic analysis. The topic of the manuscript fits with the aims and scope of the journal and is of great interest in both industry and in academia. The manuscript is well presented and easy to follow. The results are interesting, and the article contains elements of novelty. The manuscript is recommended for publication after minor revision.
- Introduction: Add information about the average composition of red mud and the amount generated. Explain whether the addition of red mud to the batch anaerobic digestion system affects the composition of the produced organic fertilizer and its application.
2.3 Analytical methods: Please describe the conditions for performing all methods applied or add references where this data is provided.
- Results and Discussion:
- Statistical analysis results are missing.
- Improve the visibility of text on figures.
- Point out future phases of research that need to be conducted to further improve the bioprocess.
- What is the cost-effectiveness of the proposed anaerobic digestion? What are the possibilities for its industrialization?
